# Malaria and dengue fever in febrile children entering healthcare facilities in Mwanza, Tanzania

Neema M. Kayange[1,2]*, Oliver Ombeva Malande[3,4,5], Philip Koliopoulos[6], Stephan Gehring[6], Britta Groendahl[6], Bahati Wajanga[7], Bahati Msaki[8], Baraka Revocatus[9], Stephen E. Mshana[10]

1 Department of Pediatrics, Weill Bugando School of Medicine, Catholic University of Health and Allied Sciences, Mwanza, United Republic of Tanzania, 2 Department of Pediatric and Child Health Bugando Medical Centre, Mwanza, United Republic of Tanzania, 3 East Africa Centre for Vaccines and Immunization (ECAVI), Kampala, Uganda, 4 Department of Pediatrics & Child Health, Makerere University, Kampala, Uganda, 5 Department of Pediatrics & Child Health, Moi University, Eldoret, Kenya, 6 Department of Pediatrics, University Medical Center of the Johannes Gutenberg University Mainz, Mainz, Germany, 7 Department of Internal Medicine, Weill Bugando School of Medicine, Catholic University of Health and Allied Sciences, Mwanza, United Republic of Tanzania, 8 Department of Pediatrics, Sekou Toure Regional Hospital, Mwanza, United Republic of Tanzania, 9 Department of Data and Statistics, Bugando Medical Centre, Mwanza, United Republic of Tanzania, 10 Department of Microbiology and Immunology, Weill Bugando School of Medicine, Catholic University of Health and Allied Sciences, Mwanza, United Republic of Tanzania

* Neemakayange@gmail.com

**Data Availability Statement:** All relevant data are within the manuscript and its Supporting information files.

## Abstract

*Plasmodium spp.* infections and cases of malaria are a long-standing public health problem for children living in middle- and low-income countries. Dengue virus causes an emerging under-recognized disease burden. A cross sectional study was conducted between March 2020 and December 2021 to determine the status of malaria and dengue fever, and the associated factors in children living in Mwanza, Tanzania. Clinical features were recorded; blood samples were analyzed using dengue NS1 rapid diagnostics test (NS1-RDT), malaria rapid diagnostic test (MRDT) and PCR and microscopy for malaria parasites. Descriptive analysis was based on infection status; odds ratio and confidence interval were used to determine the factors associated with dengue fever and malaria. The prevalence of malaria in the 436 children included in the final analysis was 15.6%, 8.5%, and 12.1% as determined by MRDT, blood smear examination and PCR, respectively. The prevalence of dengue fever determined by the NS1-RDT was 7.8%. Body rash, muscle and joint/bone pain were associated with a positive rapid dengue test result. Retro-orbital pain characterized *Plasmodium spp.* and dengue virus co-infections. Clinical signs and symptoms could not readily differentiate between malaria and dengue fever patients or patients co-infected with both causative agents underscoring the urgent need for the accurate laboratory diagnostics. Additional large-scale studies are required to assess the epidemiological burden of acute febrile illness in developing countries and to produce data that will guide empirical treatment.

**Funding:** This study was supported by the European Virus Archive goes Global project, which received funding from the European Union's Horizon 2020 research and innovation program under grant agreement No 653316. The project was partially funded by the Else-Kröner Fresenius Stiftung Klinikpartnerschaften (Bad Homburg, Germany) grant No 1601079. The funders had no role in study design, data collection and analysis, decision to publish, or preparation of this manuscript.

**Competing interests:** The authors have declared that no competing interests exist.

## Introduction

Acute febrile diseases transmitted by mosquitos cause high rates of morbidity and mortality in children living in Sub-Saharan Africa (SSA) [1]. The World Health Organization (WHO) reported 247 million cases and 619,000 deaths from *Plasmodium* infections, *P. falciparum* and *P. vivax* primarily in 2021 [1]. Approximately 96% of these deaths occurred in Africa, 80% were in children less than 5 years of age [1, 2].

Increased awareness and the introduction of artemisinin-based combined therapy have led to a significant decline in cases of malaria in recent years [3, 4]. Concurrently, an increased number of malarial-like diseases caused by viral and bacterial pathogens have been documented [5, 6]. Certain arboviruses such as dengue virus caused huge epidemic outbreaks in SAA [7, 8]. Dengue virus is a mosquito borne pathogen that is a significant global health threat affecting 2.4 billion people worldwide [5]. Outbreaks reported in Tanzania in 2014 and 2019 resulted in 7,935 cases and 13 death primarily due to dengue virus serotypes 1, 2 and 3 [9, 10]. Most patients present with either no or mild flu-like symptoms, a small percentage of cases progress in severity.

The cumulative effects of dengue virus and *Plasmodium* infections have increased in recent years with frequent outbreaks reported in several parts of Africa [11, 12]. Malaria and dengue fever share endemicity and clinical presentations; with fever being the most common symptom. Each disease has distinguishing features, e.g., episodic periods of fever in cases of malaria, hemorrhage and depletion of platelet counts in dengue fever cases [7, 13]. Due to the similarity in initial symptoms and overlapping endemicity, most of clinicians often rely on a diagnosis of *Plasmodium spp*. as the cause of disease; this results in the under diagnosis of other causative agents, i.e., dengue virus [14].

Despite similarities in presentation, malaria and dengue fever are treated differently. Malaria is treated with antimalarial drugs with no drugs available to treat dengue fever [15]. For dengue fever, clinicians rely solely on supportive treatment, even though control and preventive measures have markedly improved [16, 17]. The main strategies for the prevention and control of *Plasmodium* and dengue virus infections consist of detection and case management [18, 19]. New vaccines are available for malaria and dengue virus, several others are in development [20, 21]. Delays in diagnosing or initiating therapy for either infection can result in catastrophic outcomes. A recent study reported that the COVID-19 pandemic had a negative impact on the control of malaria in western Africa [22]. As a result, the number of cases and the mortality associated with malaria increased globally. Indeed, more cases in Western Africa were reported pre-pandemic, 2020 [22].

The prevalence of dengue fever found in studies conducted in Tanzania ranges from 0% in a study of adults and children living in Northern Tanzania, to 38.2% in a study of a pediatric population residing in Kilosa, Tanzania [5, 23]. Morbidity and mortality due to malaria remain a major health problem among children in Africa [1, 2, 24–27]. Co-existence of malaria and dengue fever in these children occurs as a consequence of diversity and changing climate. The current study was undertaken to assess the infection status and factors associated with dengue fever and malaria in children living in Mwanza, Tanzania.

## Material and methods

### Study duration and sites

The study was conducted between March 2020 and December 2021 at five sites. The period, March 2020 –December 2021 correlates with the seasonality of the diseases. Malaria tends to be more prevalent during warmer seasons while dengue fever tends to be more prevalent

during rainy seasons. Four study sites were located in urban (BMC, STRRH, BHC and NDH) and one site (SDDH) was located in rural.

## Study design and participants

This crossectional study included children ranging from 1 year to ≤12-years-of-age. This range was implemented in order to focus on two categories: children under five and those above five years-of-age, and the hospitals they attended. This range of ages covers a broad spectrum of paediatric patients and will lead to a better understanding of the most affected group(s) within the cohort. We included children who presented with acute high-grade fever (>38°C) and met the WHO criteria for a presumptive *Plasmodium* spp. or dengue virus infection [2, 15] with at least one of the following symptom present: 1) fever lasting less than 7 days, 2) vomiting, 3) headache, 4) rash or 5) joint pain. All participants enrolled at BMC were inpatients; all participants at the other four sites (STRRH, BHC, NDH, and SDDH) were outpatients. The inpatients selected in tertiary hospitals followed the same inclusion criteria and were enrolled to compare the outcomes. Patients with incomplete data and those in critical condition who required intensive critical care, e.g., trauma or acute injury, were excluded from the study (Fig 1).

The sample sizes were calculated using the Kish Leslie formula where the prevalences 38.2% and 19% for dengue fever and malaria, respectively, were used. The double proportion formula was used to calculate the associated risk factors [23, 28].

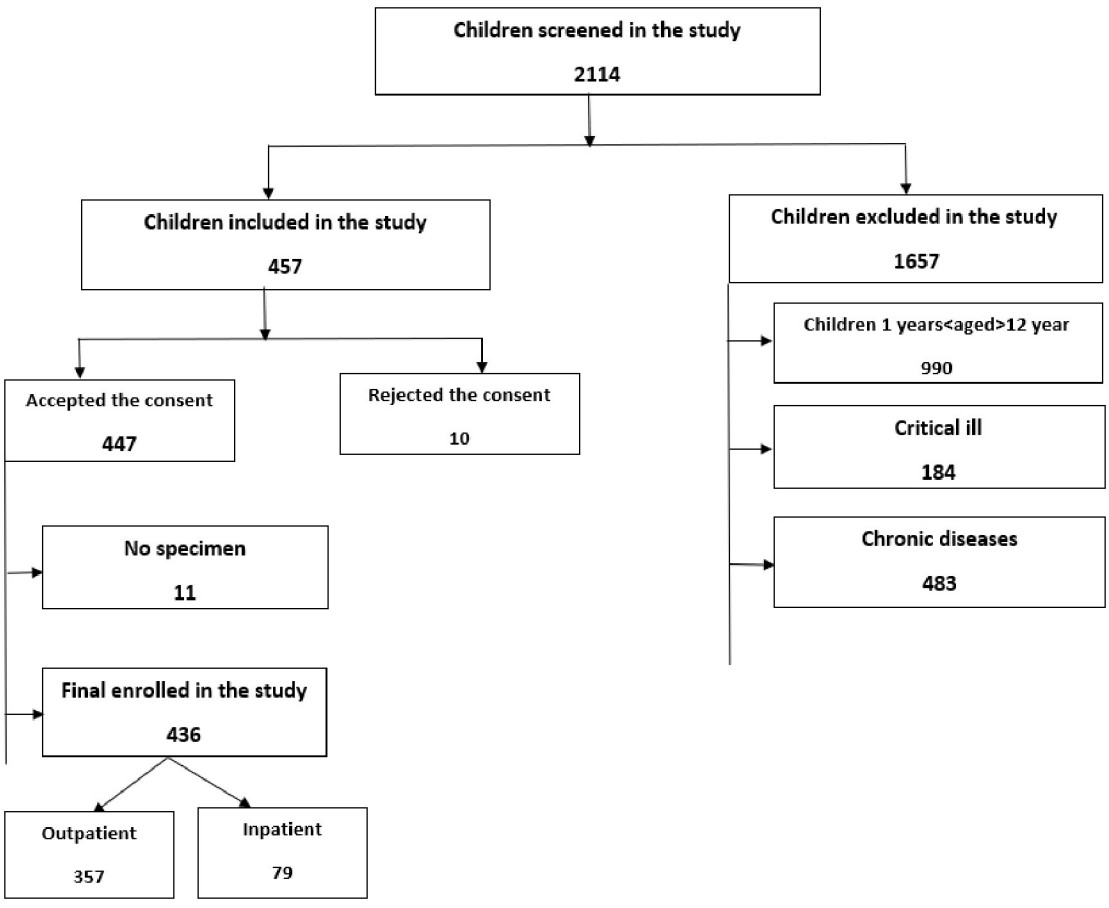

**Fig 1. Flow chart for the enrollment of study participants.**

## Sample collection and processing

Research teams composed of nurses, paediatricians and laboratory technicians who were trained in the study protocol were recruited at each facility at the beginning. Demographic and clinical information was obtained using a structured data collection form for all children who fulfilled the inclusion criteria and consented to participate. Detailed anthropometric data, i.e., height and weight, were also recorded. Outpatient medication and in-patient outcomes such as duration of hospital stay and death were recorded in the questionnaire.

Venous blood (5 ml) was collected from each study participant. Complete blood counts for all samples obtained at the five sites were performed using a Mindray haematology machine at the BMC laboratory [29]. Uncoagulated whole blood samples were mixed, the samples were processed by machine, and the results were analysed.

## Diagnostic procedures for dengue virus and *Plasmodium spp*. infections

Approximately 0.5 ml was used onsite to conduct malaria rapid diagnostic tests (MRDT, NADAL® Malaria Pf/Pan Ag 4 Species Test; nal von minden GmbH, Regensburg, Germany) and dengue rapid IgM/IgG tests as reported in previous publications [28, 30]. Dengue was diagnosed using the SD Bioline NSI IgM/IgG combo rapid test according to the manufacturer's instructions (Standard Diagnostics, Inc., Suwon City, Korea) [28, 31]. Past infection was indicated by a dengue-positive rapid IgG test.

An additional 2.5 ml of blood was centrifuged and stored at -20˚C then transferred to the BMC laboratory for a full blood picture. Thick blood smears were prepared, stained with 10% Giemsa and examined microscopically to detect malaria parasites as described [32]. The remaining blood sample was spotted onto Whatman 903 Protein Saver filter cards. Serum samples and filters cards were transported to the Catholic University of Health and Allied Sciences then to Mainz, Germany where viral RNA was extracted as previously described using a High Pure Viral Nucleic Acid Kit (Roche Diagnostic, Mannheim, Germany) according to the manufacturer's recommendations [28]. Subsequently, a multiplex reverse-transcriptase-polymerase-chain-reaction (MPCR)-ELISA for malaria-like diseases was performed at Children's Hospital in Mainz and the Fraunhofer Institute in Leipzig, Germany [28]. This assay covered a panel of nine mosquito-transmitted diseases and included a highly sensitive primer set that targets *Plasmodium spp*. 18S rRNA gene (Table 1).

Nucleic acid was extracted and purified from pathogens in a biosafety level 2 cabinet following strict biosecurity standards. Each series of 9 specimens included one negative control

**Table 1. Pathogens detected by MPCR[a].**

| Panel No | Pathogen |
|---|---|
| 1 | Dengue virus |
| 2 | West Nile virus |
| 3 | Zika virus |
| 4 | Yellow Fever virus |
| 5 | Semliki Forest virus |
| 6 | O´nyong' nyong virus |
| 7 | Chikungunya virus |
| 8 | Rift Valley fever virus |
| 9 | *P. falciparum*, *P. vivax*, *P. malariae* |

[a]Multiplex reverse transcriptase polymerase chain reaction.

(NaCl) to monitor cross-contamination. Preparation of PCR reagents and nucleic acid extraction were conducted in different rooms. Gene amplification and gel electrophoresis were performed as previously described [28]. The PCR product was incubated with 3′-biotinylated capture probe and anti-digoxigenin-peroxidase (Roche Diagnostics) on streptavidin-coated microtiter plates. Samples were classified based upon an optical density cut-off value of 0.4 $OD_{405}$; values between 0.2 and 0.4 were considered borderline and retested by single-plex PCR before classification. All *Plasmodium* spp.-positive samples were retested by trivalent PCR to differentiate between *P. falciparum*, *P. vivax* and *P. malariae*.

## Data analysis

Data were double entered into Microsoft excel and analyzed using STATA version 15 [33]. Results were summarized using proportions (%) for categorical data and mean ± SD or medians (IQR) for continuous variables. Categorical variables were compared using Pearson's chi–square or Fisher's exact test where appropriate. Univariate followed by multivariate logistic and Poisson regression analyses were performed to determine positive predictors for dengue fever and malaria, respectively. Predictors that were investigated included: socio-demographic factors, clinical features, and laboratory parameters. Odds ratios (OR) and prevalence ratio (PR) with 95% confidence intervals (CI) are reported. Predictors with *P* values <0.05 were considered statistically significant for both univariate and multivariate analyses. The primary analysis-measure was to assess the prevalence of two outcomes; the odds ratio and prevalence ratio were used to assess the associated factors in dengue and malaria within the study population, respectively.

## Ethical approval and informed consent

Ethical approval was obtained from the National Institute of Medical Research of Tanzania (NIMR/HQ/R.8a/Vol.IX/2641). Ethic clearance was also obtained from the joint Catholic University of Health and Allied Sciences/ Bugando Medical Centre Ethics and Review Committee (CREC/455/2020). Permission was obtained from the administrations of the Bugando Medical Centre (BMC), Sekou Toure Region Hospital (STRRH), Buzuruga Health Centre (BHC), Nyamagana District Hospital (NDH) and Sengerema District Hospital (SDDH). Written informed consent was obtained from a parent or guardian of each participating child. All participants diagnosed with malaria or dengue fever were treated according to the WHO guidelines [15, 17].

## Results

### Social demographic description of the study population

Four hundred thirty-six febrile children ranging in age from 1 to 12 years-old were enrolled (BHC, 86; NDH, 85; SYRRH, 70; SDDH, 101; BMC, 94). Of these children, 79.5% were outpatients and 18.1% were inpatients. There were more males (56.7%) than females (43.3%); the mean age of all children was 1.9 years-old with standard deviation of 1.7 years-old. Most of the children slept under protected bed nets (98.3%); slightly fewer than half lived in an environment contaminated with stagnant water (Table 2).

### Prevalence and clinical manifestations of *Plasmodium* spp. and dengue virus infections

All 436 children enrolled in this study presented with an axillary temperature higher than 37.5˚C. The highest temperature observed was 41˚C, the average was 38.6˚C. One hundred

**Table 2. Background/socio-demographic characteristics of study participants.**

| Variable | N | % |
|---|---|---|
| **Study site[b]** | | |
| BHC | 86 | 19.7 |
| NDDH | 85 | 19.5 |
| STTRH | 70 | 16.0 |
| SDDH | 101 | 23.2 |
| BMC | 94 | 21.6 |
| **Sex** | | |
| Male | 245 | 56.2 |
| Female | 191 | 43.8 |
| **Age (years)** | | |
| Mean ± SD | 1.9 ± 1.7 | |
| 1–5 | 407 | 93.3 |
| 6–11 | 29 | 6.7 |
| **Mosquito net use** | | |
| Yes | 410 | 98.3 |
| No | 7 | 1.7 |
| **Insecticide use** | | |
| Yes | 43 | 9.9 |
| No | 393 | 90.1 |
| **Stagnant water** | | |
| Yes | 192 | 44.0 |
| No | 244 | 56.0 |

[a]Number (N) and percentage (%) of children in each group are indicated.

[b]BMC, Bugando Medical Centre; STRRH, Sekou Toure Region Referral Hospital; BHC, Buzuruga Health Centre; NDH, Nyamagana District Hospital and SDDH, Sengerema Designated District Hospital.

forty-seven children (34%) were vomiting, 123 (28%) had diarrhea and 25 (6%) had body rashes. Thirty-four (8%) of the children were diagnosed with dengue virus infections by the NS1-RDT; 29 (7%), 5 (1%) and 0 were diagnosed by demonstrating anti-dengue serum IgG, anti-dengue serum IgM and PCR, respectively. A significantly greater percentages of dengue fever-positive than fever-negative children exhibited rashes, and retroorbital, muscle and joint/bone pain (Table 3). All other arboviral diseases tested in the Table 1 panel by MPCR were negative.

Sixth-eight cases of malaria (15.6%) were identified by MRDT, compared to 36 cases (8.3%) identified by visual examination of thick blood smears and 32 (12.1%) were determined by PCR. Greater percentages of malaria-positive than -negative children exhibited headache and retroorbital pain, albeit neither difference attained statistical significance (Table 4). Notably, of the 436 children enrolled, 3.4% (15/436) were coinfected with *Plasmodium spp*. and dengue virus. Multivariate analysis determined that retroorbital pain was often symptomatic of these coinfected individuals (OR 3.72, 95% CI 2.10–6.59, $P$ <0.001).

## Factors affecting dengue virus and *Plasmodium* spp. infections

Univariate analysis of the factors that potentially affect dengue rapid test-positive children enrolled in the study determined that the lack of household education, severe malnutrition and the infrequent use of mosquito nets increased the risk of dengue virus infection; neither

**Table 3. Clinical spectrum of dengue fever in febrile children visiting healthcare facilities in Mwanza, Tanzania[a].**

| Symptom | Dengue fever | | Total | P value |
|---|---|---|---|---|
| | Positive[b] | Negative | | |
| **Fever duration (days)** | | | | |
| 1–3 | 28 (7.8) | 333 (92.2) | 361 | |
| 4–7 | 6 (8.0) | 69 (92.0) | 75 | 0.943 |
| **Vomiting** | | | | |
| Yes | 14 (9.5) | 133 (90.5) | 147 | 0.338 |
| No | 20 (6.9) | 269 (93.1) | 289 | |
| **Diarrhea** | | | | |
| Yes | 13 (10.6) | 110 (89.4) | 123 | 0.176 |
| No | 21 (6.7) | 292 (93.3) | 313 | |
| **Body rash** | | | | |
| Yes | 8 (32.0) | 17 (68.0) | 25 | 0.001[c] |
| No | 26 (6.3) | 385 (93.7) | 411 | |
| **Headache** | | | | |
| Yes | 4 (12.1) | 29 (87.9) | 33 | 0.335 |
| No | 30 (7.5) | 372 (92.5) | 402 | |
| **Retroorbital pain[c]** | | | | |
| Yes | 2 (66.7) | 1 (33.3) | 3 | 0.017 |
| No | 32 (7.4) | 400 (92.6) | 432 | |
| **Muscle pain[c]** | | | | |
| Yes | 8 (61.5) | 5 (38.5) | 13 | 0.001 |
| No | 26 (6.2) | 397 (93.8) | 423 | |
| **Joint/bone pain** | | | | |
| Yes | 4 (40.0) | 6 (60.0) | 10 | 0.005 |
| No | 30 (7.0) | 396 (93.0) | 426 | |

[a]Number and (percentage) of children in each group are given.

[b]NS1-RDT-positive.

[c]Percentages of dengue fever-positive and -negative patients exhibiting symptom are significantly different, Pearson's chi-square and Fisher's exact tests.

sex nor age was a factor (Table 5). Similar analysis was done it was found out Neither sex nor age was associated with positive *Plasmodium spp* (Table 6). Multivariate analysis of these same factors determined that the use of mosquito nets in the case of dengue virus, and vaccination and the use of mosquito nets in the case of *Plasmodium spp*. inhibited infection independent of other variables.

## Discussion

The prevalence of malaria among children visiting five healthcare facilities in Mwanza, Tanzania was determined to be 15.3%, 8.3%, and 12.1% by MRDT, microscopy and PCR of *Plasmodium spp*. DNA, respectively. The overall prevalence of both presumptive and prior exposure to dengue virus in children was 7.8%; the prevalence of *Plasmodium spp*. and dengue virus coinfections was 3.4%. Malaria and dengue fever shared common clinical presentations, fever and muscle pain; additionally, coinfected patients experienced retro-orbital pain.

The prevalence of malaria among children enrolled in the present study was less than that found in children in previous studies conducted between 2016 and 2018, i.e., prior to the COVID-19 pandemic, when the prevalence assessed by PCR was 20% [28, 34]. The ongoing

**Table 4. Clinical spectrum of malaria in febrile children attending healthcare facilities in Mwanza, Tanzania[a].**

| Symptom | Malaria | | Total | P value |
|---|---|---|---|---|
| | Positive[b] | Negative | | |
| **Fever duration (days)** | | | | |
| 1–3 | 53 (14.7) | 308 (85.3) | 361 | 0.293 |
| 4–7 | 15 (20.0) | 60 (80.0) | 75 | |
| **Vomiting** | | | | |
| yes | 23 (15.7) | 124 (84.3) | 147 | 0.984 |
| no | 45 (15.6) | 244 (84.4) | 289 | |
| **Diarrhea** | | | | |
| yes | 21 (17.1) | 102 (82.9) | 123 | 0.594 |
| no | 47 (15.0) | 266 (85.0) | 313 | |
| **Rash** | | | | |
| yes | 3 (12.0) | 22 (88.0) | 25 | 0.781 |
| no | 65 (15.8) | 346 (84.2) | 411 | |
| **Headache** | | | | |
| yes | 24 (72.7) | 9 (27.3) | 33 | 0.054 |
| no | 344 (85.4) | 59 (14.6) | 403 | |
| **Retroorbital pain** | | | | |
| yes | 2 (66.7) | 1 (33.3) | 3 | 0.065 |
| no | 66 (15.2) | 367 (84.8) | 433 | |
| **Muscle pain** | | | | |
| yes | 4 (30.8) | 9 (69.2) | 13 | 0.129 |
| no | 64 (15.1) | 359 (84.9} | 423 | |
| **Joint/bone pain[c]** | | | | |
| yes | 1 (10.0) | 9 (90.0) | 10 | 0.029 |
| no | 67 (15.7) | 359 (84.3) | 426 | |

[a]Number and (percentage) of children in each group are provided.

[b]MRDT-positive.

[c]Percentage of malaria-positive and -negative patients expressing joint pain is significantly different, Pearson's chi-square and Fisher's exact tests.

effort of the Tanzania Government and the Tanzania National Malaria Control Program, as well as the test, treat and track recommendation by the WHO, may account for this difference. However, data collection reported here during both dry and rainy seasons when the prevalence is slightly higher in the latter could be a contributing factor. Notably, the testing methods used to determine the prevalence of malaria, e.g., MRDT versus visual examination of Giemsa-stained blood smears, is a matter of debate and undoubtedly an added factor that contributes to this discrepancy. The accuracy of blood smear microscopy depends upon the examiner's skill, which requires regulation by the WHO in order to optimize the accuracy of the test [4, 34]. Microscopic examination of Giemsa-stained thick blood smear remains the standard diagnostic test for acute malaria infections in many areas. Previous documentation found the sensitivity to be 33% [3, 35]. This poses a particularly difficult challenge identifying low-density parasitemia, dependent upon the examiner's skill, a major limitation. Regardless, the prevalence reported here was higher than the 5% of 0- to 5-year-old children in Mwanza reported previously in 2022 Tanzania demographic health survey [36]. An ongoing effort to reduce malaria and prevent infections in children should be a priority for both the Tanzanian government and the WHO.

**Table 5. Factors affecting dengue virus infection of febrile children presenting at healthcare facilities in Mwanza, Tanzania.**

| Factor | Univariate analysis | | | Multivariate analysis | | |
|---|---|---|---|---|---|---|
| | OR | 95% CI | *P* value | OR | 95% CI | *P* value |
| **Sex** | | | | | | |
| Female | 0.82 | (0.40, 1.71) | 0.606 | 0.91 | (0.43,1.90) | 0.798 |
| Male | 1.00 | | | 1.00 | | |
| **Age (years)** | | | | | | |
| 1–5 | 0.48 | (0.13, 1.74) | 0.265 | 0.67 | (0.17,2.61) | 0.563 |
| 6–11 | 1.00 | | | 1.00 | | |
| **Household education level** | | | | | | |
| primary | 0.21 | (0.07,0.65) | 0.007 | 0.27 | (0.07, 098) | 0.048 |
| secondary | 0.11 | (0.03,0.41) | 0.001 | 0.12 | (0.03,0.52) | 0.005 |
| college/university | 0.07 | (0.01,0.68) | 0.021 | 0.06 | (0.01,063) | 0.020 |
| none/incomplete | 1.00 | | | | | |
| **Nutrition** | | | | | | |
| mildly malnourished | 0.96 | (0.21, 4.29) | 0.955 | 1.07 | (0.31,3.76) | 0.914 |
| moderately malnourished | 2.46 | (0.31, 19.78) | 0.392 | 1.26 | (0.26,6.23) | 0.777 |
| severely malnourished | 12.00 | (1.52, 95.02) | 0.019 | 2.65 | (0.30,23.55) | 0.383 |
| nourished | 1.00 | | 1.00 | | | |
| **Mosquito net use** | | | | | | |
| yes | 0.13 | (0.05, 0.35) | 0.001 | 0.16 | (0.06,0.46) | 0.001 |
| no | 1.00 | | | 1.00 | | |
| **Insecticide use** | | | | | | |
| yes | 2.64 | (1.07,6.48) | 0.035 | 1.76 | (0.65,4.74) | 0.266 |
| no | 1.00 | | | 1.00 | | |
| **Stagnant water** | | | | | | |
| yes | 0.58 | (0.28,1.23) | 0.157 | 0.69 | (0.32,1.50) | 0.346 |
| no | 1.00 | | | 1.00 | | |

OR, odds ratio; CI, confidence interval; *P* value, logistic regression.

Presumptive acute anti-dengue IgM and IgG reflecting prior exposure were detected in 1.2% and 6.7%, respectively, of individuals enrolled in the current study. These results are very low compared to a prevalence of 38.2% reported by a study performed in Kilosa, Tanzania [23]. The latter was a cross-sectional study of a pediatric, outpatient population conducted in a coastal region. Both the geographical zone and method of detection (ELISA) differed significantly from the current study. None of the participants enrolled at all five sites reported here tested positive for dengue by PCR. Conceivably, this is due to the fact that most of the viral RNA is detected in patients between 0 and 7 days following the onset of symptoms, which approximates the duration of fever in most viral infections [23, 37]. Therefore, the possibility exists that the low percentage of IgM positive, PCR negative patients was due to the collection of samples near the end of the acute phase of infection. This finding emphasizes the importance of initiating ongoing surveillance of arboviruses, especially dengue, where all dengue virus serotypes were isolated in Tanzania during outbreaks [9, 10].

The results shown here document the similarities in the clinical presentations of presumptive dengue fever and confirmed malaria cases. Children co-infected with *Plasmodium* spp. and dengue virus were three times more likely to experience retro-orbital pain than children with malaria only. The clinical presentation of fever, headache and muscle pain correlate with

**Table 6. Factors affecting *Plasmodium spp*. infections of children presenting with fever at five healthcare facilities in Mwanza, Tanzania.**

| Variable | Univariate analysis | | | Multivariate analysis | | |
|---|---|---|---|---|---|---|
| | PR | 95% CI | *P* value | PR | 95% CI | *P* value |
| **Sex** | | | | | | |
| female | 1.41 | (0.91, 2.19) | 0.126 | 1.60 | (1.01,2.55) | 0.044 |
| male | 1.00 | | | 1.00 | | |
| **Age (Years)** | | | | | | |
| 1–5 | 0.64 | (0.29,1.42) | 0.269 | 0.84 | (0.337,2.13) | 0.716 |
| 6–12 | 1.00 | | | 1.00 | | |
| **Marital Status** | | | | | | |
| monogamy | 0.40 | (0.21,1.17) | 0.75 | 0.39 | (0.19, 0.81) | 0.011 |
| polygamy | 0.49 | (0.17,1.43) | 0.045 | 0.42 | (0.14,1.33) | 0.142 |
| cohabiting | 0.54 | (0.14,2.14) | 0.383 | 0.62 | (0.16,2.47) | 0.499 |
| separate | 0.23 | (0.03,1.62) | 0.139 | 0.01 | (0.001,0.003) | 0.0001 |
| widowed | 1.00 | | | 0.29 | (0.04,2.43) | 0.236 |
| single | 1.00 | | | 1.00 | | |
| **Vaccination** | | | | | | |
| yes | 0.30 | (0.15, 0.57) | 0.001 | 0.32 | (0.14,0.72) | 0.006 |
| no | 1.00 | | | 1.00 | | |
| **Mosquito net use** | | | | | | |
| yes | 0.35 | (0.20, 0.63) | 0.001 | 0.73 | (0.37,1.43) | 0.363 |
| no | 1.00 | | | 1.00 | | |
| **Headache** | | | | | | |
| yes | 1.86 | (1.02, 3.41) | 0.044 | 1.35 | (0.62,2.92) | 0.447 |
| no | 1.00 | | | 1.00 | | |
| **Retroorbital pain** | | | | | | |
| yes | 4.37 | (1.91, 10.03) | 0.001 | 4.94 | (2.28,10.72) | 0.001 |
| no | 1.00 | | | 1.00 | | |

PR, prevalence ratio; CI, confidence interval; *P* value, Poisson regression

the findings of other studies conducted in Africa [9, 34]. The lack of unique, clinical manifestation poses a diagnostic challenge in countries where resources are limited.

Univariate analysis determined that children suffering severe malnutrition were 12-times more likely to be infected with dengue virus than those who were nourished. Immunosuppression, typical of underweight children, may be a contributing factor. Multivariate analysis indicated, however, that severe malnutrition did not promote dengue virus infection independent of the other factors assessed. Multivariate analysis found, though, that the level of household education and the use of mosquito nets reduced dengue virus infection independent of these factors. Similarly, multivariate analysis found that vaccination and the use of mosquito nets significantly reduced *Plasmodium* spp. infections independent of these additional factors. Co-endemicity of other infectious agents in the study area undoubtedly contributes to the lack of significant correlation between some environmental factors measured and *Plasmodium* spp. and dengue virus infections. Additional studies are needed to clarify these relationships.

*Plasmodium spp*. and dengue virus co-infections occurred in ~3.4% of children. Systematic review and meta-analysis studies conducted in Africa reported an increase in concurrent infections; East Africa and Tanzania were among the countries reporting [37, 38]. The prevalence of malaria, and concurrent cases of malaria and dengue fever occurred more often in children

than adults. Children experience greater exposure to mosquito bites for longer periods of time; their relatively immature immune systems render them more susceptible to mosquito borne illnesses [39–41].

While the prevalence of presumptive dengue fever cases determined serologically in the present study was low, the presence of anti-dengue IgM and IgG in malaria-positive cases suggests a significant chance that the symptoms in 7% percent of malaria cases involved dengue virus infection. Dengue virus infections are still considered uncommon despite serological evidence suggesting otherwise. Most clinicians rarely consider dengue virus, a relatively new phenomenon, when diagnosing febrile illnesses since diseases such as malaria, pediatric HIV, and typhoid fever are endemic and cause high rates of pediatric mortality in most SSA countries. In the study reported here, 10% of children had malaria confirmed by microscopy and PCR, others had dengue fever or an undiagnosed cause of febrile illness. Others causes of febrile illnesses such as enteric fever, urinary tract infections, and viruses, e.g., chikungunya, West Nile and zika, should also be considered although not the focus of the current evaluation. Overlapping clinical symptoms and the limited diagnostic capacity of most hospitals in middle- and low-income settings are major challenges. As a result, most clinicians overlook dengue virus, as well as other causes of acute febrile illness, leading to misdiagnoses. This lack of clinical suspicion of other causes contributes to the imminent outbreaks of dengue and other febrile illnesses without sufficient warning or public health preparedness.

*Plasmodium spp*. and dengue virus infections are transmitted by mosquitoes, Anopheles and Aedes, respectively. Environmental conditions like stagnant water in holding containers like buckets provide breeding conditions that impact the rate of transmission. The Tanzania government and the malaria control program must continue to promote the availability of protective mosquito nets and encourage parents and caregiver to assure their children sleep under protective nets to reduce *Plasmodium spp*. transmission. Indeed, confirmed cases of malaria still occurred in 10% of children reported herein. Notably, bed nets are less effective in preventing dengue virus than *P. malariae* infections due to the day biting habit of Aedes mosquitoes. A study conducted in Dar es Salama found that Aedes aegypti mosquitoes accounted for 16.8% of those recovered in water holding containers, but none of 763 females tested positive for dengue virus by RT–PCR [42]. These results emphasize the need for the Tanzania government to continue mosquito vector surveillance in order to inhibit the transmission of malaria and the probability of future dengue fever outbreaks.

The study described herein has several limitations. 1) The SD Bioline NSI IgM/IgG combo rapid test cross-reacts extensively with other arboviruses, e.g., chikungunya and Zika. Despite this limitation, the rapid test is useful in countries where resources are limited. Clinicians can quickly confirm diagnoses and optimize case management, avoiding unnecessary consumption of antimalarial or antibiotics. 2) Some samples were collected 5 or more days after the onset of fever. This negates relating dengue virus infections or dengue virus/*Plasmodium* spp. co-infections to other clinical characteristics. However, the findings of the present study can be used as a basis for further large-scale hospital and community exploration of the etiologies of febrile illnesses in regions of Tanzania. Assessing the outcomes and complications of *Plasmodium spp*. and dengue virus, mono- or co-infections, through longitudinal studies of these patients is recommended.

In conclusion, this study demonstrates different prevalence patterns for *Plasmodium* spp. and dengue virus infecting children living in Mwanza, Tanzania. Although the PCR results for dengue virus infection were negative, serological evidence indicates the presence of dengue virus and highlights the need for urgent action to combat both diseases. Dengue and other arbovirus are overlooked in Tanzania due to the hyperendemicity of malaria. The cause of febrile illness in ~80% of the children, however, was neither dengue virus nor *Plasmodium*

spp. highlighting the need to look for other causative agents. Combined measures that include vector control, community-sensitization, and improved diagnostics must be promoted to effectively combat such febrile diseases.

## Supporting information

**S1 Dataset.**
(ZIP)

## Acknowledgments

The authors want to thank the Weill Bugando School of Medicine of the Catholic University of Health and Allied Sciences, Mwanza, Tanzania. We appreciate the hospital administration at Bugando Medical Centre Mwanza, Tanzania, and the Department of Pediatrics at the University Medical Center of Johannes Gutenberg University Mainz, Germany. We appreciate the support provided by the Samuel Wood Library/C.V. Starr Biomedical Information Center, Weill Cornell Medical College, New York, United States of America. We also appreciate the support provided by the Mwanza region commissioner and district medical officers of the Ilemela and Nyamagana districts. We also extend our gratitude to the medical officers in charges and staff of Buzuruga health center, Nyamagana district hospital, Sekou Toure Regional Referral hospital, Sengerema District hospital and Bugando Medical Centre. Additionally, we would like to thank the parents and children for their voluntary participation in this study. We are grateful for the research team, Dr. Gayo Mpalala, Mr. Peter Nandi, Mr. Frank Elias. Special thanks to Stephen H. Gregory (Providence, RI, USA) for his help writing, editing, and reviewing this manuscript, and to Scott Bailey (Dallas, TX, USA) for providing statistical support.

## Author Contributions

**Conceptualization:** Neema M. Kayange, Stephan Gehring, Stephen E. Mshana.

**Data curation:** Neema M. Kayange, Oliver Ombeva Malande, Bahati Wajanga, Baraka Revocatus, Stephen E. Mshana.

**Formal analysis:** Neema M. Kayange, Stephan Gehring, Baraka Revocatus.

**Funding acquisition:** Philip Koliopoulos, Stephan Gehring, Britta Groendahl.

**Investigation:** Neema M. Kayange, Britta Groendahl, Baraka Revocatus, Stephen E. Mshana.

**Methodology:** Neema M. Kayange, Oliver Ombeva Malande, Philip Koliopoulos, Britta Groendahl, Baraka Revocatus, Stephen E. Mshana.

**Project administration:** Neema M. Kayange, Stephan Gehring, Stephen E. Mshana.

**Resources:** Stephan Gehring.

**Supervision:** Oliver Ombeva Malande, Stephan Gehring, Stephen E. Mshana.

**Validation:** Neema M. Kayange, Philip Koliopoulos, Stephan Gehring, Britta Groendahl, Bahati Wajanga, Bahati Msaki, Baraka Revocatus, Stephen E. Mshana.

**Visualization:** Neema M. Kayange, Bahati Wajanga, Bahati Msaki.

**Writing – original draft:** Neema M. Kayange, Oliver Ombeva Malande, Philip Koliopoulos, Stephan Gehring, Britta Groendahl, Bahati Wajanga, Baraka Revocatus, Stephen E. Mshana.

**Writing – review & editing:** Neema M. Kayange, Oliver Ombeva Malande, Philip Koliopoulos, Stephan Gehring, Britta Groendahl, Bahati Wajanga, Bahati Msaki, Baraka Revocatus, Stephen E. Mshana.

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
