## [Decision Letter · Decision Letter 0]

28 May 2024

PONE-D-24-14812Malaria and dengue fever in febrile children entering healthcare facilities in Mwanza, TanzaniaPLOS ONE

Dear Dr. Kayange,

Thank you for submitting your manuscript to PLOS ONE. After careful consideration, we feel that it has merit but does not fully meet PLOS ONE’s publication criteria as it currently stands. Therefore, we invite you to submit a revised version of the manuscript that addresses the points raised during the review process. Kindly address all points raised by the reviewers in a point by point detailed reply, indicating what changes have been introduced into the manuscript. Please justify if a comment has not been considered for the manuscript. 

We look forward to receiving your revised manuscript.

Kind regards,

Benedikt Ley, PhD

Academic Editor

PLOS ONE

“This study was supported by the European Virus Archive goes Global project, which received funding from the European Union’s Horizon 2020 research and innovation program under grant agreement No 653316. The project was partially funded by the Else-Kröner Fresenius Stiftung Klinikpartnerschaften (Bad Homburg, Germany) grant No 1601079.”

3. In the online submission form, you indicated that your data is available only on request from a third party. Please note that your Data Availability Statement is currently missing contact details for the third party, such as an email address or a link to where data requests can be made. Please update your statement with the missing information.

Reviewers' comments:

Reviewer's Responses to Questions

**Comments to the Author**

1. Is the manuscript technically sound, and do the data support the conclusions?

Reviewer #1: Yes

Reviewer #2: Yes

2. Has the statistical analysis been performed appropriately and rigorously? 

Reviewer #1: No

Reviewer #2: Yes

3. Have the authors made all data underlying the findings in their manuscript fully available?

Reviewer #1: Yes

Reviewer #2: Yes

4. Is the manuscript presented in an intelligible fashion and written in standard English?

Reviewer #1: Yes

Reviewer #2: Yes

5. Review Comments to the Author

Reviewer #1: The manuscript can benefit from the following comments:

1. The authors indicate that cross sectional study was conducted between March 2020 and December 2021 to determine the status of malaria and dengue fever....why did the study target the period March 2020 to December 2021? Was there something special and related to the diseases of study linked to this timeline?

2. The following statement in the introduction needs references: A recent study reported that the Covid 19 pandemic had no impact on the prevalence, morbidly or mortality associated with malaria globally.

3. The main reason why we would want to identify risk factors associated with dengue fever and malaria in children living in Mwanza, Tanzania following the COVID-19 pandemic, is not fully justified. Why did we want to establish these factors? What are the issues surrounding the risk factors linked to dengue? Are they identified and to what levels? Why would we want to identify them anyways?

4. Why were the children aged 1 to 12 years targeted for this study?

5. Was there need for a sample size calculation for this study? If not, why?

6. The data analysis section talks about 'predictors' while the objective was to identify risk factors. Are predictors and risk factors the same?

7. It is unclear how the 2 primary outcomes: prevalence and risk factors were used in the regression analyses. Were they merged or separated as dependent variables?

8. In the results, ages have been stratified into 1-5 (difference of 5) and 6-12 (difference of 7). What was the criteria for this stratification?

9. On Table 5, what was the rationale for having those groups as the reference groups for the regression analyses? The same applies to Table 6.

10. On Table 6, on the variable: insecticide use under multivariate analysis..the OR=0.95, the 95% CI=0.95-0.95 and the p-value for that variable=0.001. Mathematically, this is not possible. There is a problem with this analysis outcome.

11. Same Table 6: multivariate analysis...the p-value for age =0225.

12. I still did not get clearly what the difference was at the analytical stage between univariate and multivariate regression analysis. Was anything adjusted for and why?

Reviewer #2: The study aimed to determine the status of malaria and dengue fever, and the associated risk factors in children living in Mwanza, Tanzania. The prevalence of malaria in 436 children included in the final analysis was 15.6%, 8.5%, and 12.1% determined by MRDT, blood smear examination, and PCR, respectively. The prevalence of dengue fever determined by the NS1-RDT was 7.8%.

The study's strength was that molecular diagnosis using Polymerase Chain Reaction was incorporated into the study with a higher sensitivity. However, there are some issues with the manuscript that need to be addressed, which I have detailed here.

Introduction

Line 55: Give a reference.

Line 79: The sentence is not clear.

Line 86: Reference

Line 90: You mentioned studies in Tanzania, there should be more than one reference.

Line 93: Give a reference.

Line 94: Replace with associated factors.

Methods

Line 120: selecting inpatients could lead to selection bias. These are severely ill patients and might have been diagnosed.

Line 177: Prevalence OR

Line 180: Replace with associated factors.

Results

Line 186: Replace with Males accounted for 56.2% and the mean age .....

Line 189: Replace positive and negative with yes and no.

Line 197-198: Be consistent with percentages. Please use %.

Line 204: change all the categories to yes and no.

Line 214: Prevalence of malaria=15.6%

Prevalence of dengue= 7.8

Coinfection = 14.7%?

The denominator should be 436 i.e. 15/436 =3.4% because you said of the 436 in the sentence.

Line 218: change the positive and negative to yes and no.

Line 235: Use the legend to show that 1= = reference in the table and use yes or no instead of positive and negative.

Discussion

Line 247-248: Please check the calculation.

Line 259: Is microscopy not the gold standard?

Line 289: compared with those who are nourished i.e. the comparison group.

Line 296: after adjusting for other covariates.

Line 301: check the calculation.

6. PLOS authors have the option to publish the peer review history of their article (what does this mean?). If published, this will include your full peer review and any attached files.

Reviewer #1: **Yes: **Collins Ouma, Maseno University, Kenya

Reviewer #2: No

---

## [Author Response · Author response to Decision Letter 0]

12 Aug 2024

RESPONSE TO REVIEWERS COMMENTS FOR PONE-D-24-14812

Title: Malaria and dengue fever in febrile children entering healthcare facilities in Mwanza, Tanzania

PLOS ONE

Response to the Journal requirement and Reviewers’ comments

We thank the Reviewers for their helpful comments. Find our detailed responses to these comments below. 

Journal requirements

Comment 1.

• Our manuscript was revised in accordance with the Journal’s requirements.

“This study was supported by the European Virus Archive goes Global project, which received funding from the European Union’s Horizon 2020 research and innovation program under grant agreement No 653316. The project was partially funded by the Else-Kröner Fresenius Stiftung Klinikpartnerschaften (Bad Homburg, Germany) grant No 1601079.”

3. In the online submission form, you indicated that your data is available only on request from a third party. Please note that your Data Availability Statement is currently missing contact details for the third party, such as an email address or a link to where data requests can be made. Please update your statement with the missing information.

• The Data Availability Statement now includes third party contact information: Directorate of Research and Innovation, Catholic University of Health and Allied Sciences, PO box 1464, Mwanza, Tanzania (dmorona@bugando.ac.tz.com)

• Role of Funder Statement: “The funders had no role in study design, data collection and analysis, decision to publish, or preparation of this manuscript” (revised manuscript; page 28, lines 511-512. This statement is also included in the cover letter. 

• The map shown in Figure 1 was drawn using a free, Geographic Information System (GIS) mapping program. (revised manuscript; page 5, lines 100-101).

• The references are accurate.

Reviewer's Responses to Questions

Comments to the Author

1. Is the manuscript technically sound, and do the data support the conclusions?

Reviewer #1: Yes

Reviewer #2: Yes

2. Has the statistical analysis been performed appropriately and rigorously?

Reviewer #1: No

Reviewer #2: Yes

3. Have the authors made all data underlying the findings in their manuscript fully available?

Reviewer #1: Yes

Reviewer #2: Yes

4. Is the manuscript presented in an intelligible fashion and written in standard English?

Reviewer #1: Yes

Reviewer #2: Yes

5. Review Comments to the Author

Reviewer #1: The manuscript can benefit from the following comments:

1. The authors indicate that cross sectional study was conducted between March 2020 and December 2021 to determine the status of malaria and dengue fever....why did the study target the period March 2020 to December 2021? Was there something special and related to the diseases of study linked to this timeline?

• The period, March 2020 – December 2021 correlates with the seasonality of the diseases. Malaria tends to be more prevalent during warmer seasons while dengue fever tends to be more prevalent during rainy seasons (revised manuscript; page 5, lines 95-98). 

2. The following statement in the introduction needs references: A recent study reported that the Covid 19 pandemic had no impact on the prevalence, morbidly or mortality associated with malaria globally.

• The statement was corrected (revised manuscript; page 4, lines 81-84 and reference 22). 

3. The main reason why we would want to identify risk factors associated with dengue fever and malaria in children living in Mwanza, Tanzania following the COVID-19 pandemic, is not fully justified. Why did we want to establish these factors? What are the issues surrounding the risk factors linked to dengue? Are they identified and to what levels? Why would we want to identify them anyways?

• The sentence was corrected to state that “the Covid 19 pandemic had a negative impact on the control of malaria” (Revised manuscript; page 4, line 82).

 4. Why were the children aged 1 to 12 years targeted for this study?

• The aim of including children aged 1 to 12 years-old was to focus on two categories: children under five and those above five years-of-age, and the hospitals in which they entered. The range of ages covers a broad spectrum of paediatric patients and will lead to a better understanding of the most affected group(s) within the cohort (revised manuscript; page number 6, lines 103-107). 

5. Was there need for a sample size calculation for this study? If not, why?

• The sample sizes were calculated using the Kish and Leslie formula where the prevalences 38.2% and 19% for dengue fever and malaria, respectively, were used. The double proportion formula was used to calculate the associated risk factors (revised manuscript; page 6, lines 117-119; sample size references: 23 and 25).

6. The data analysis section talks about 'predictors' while the objective was to identify risk factors. Are predictors and risk factors the same?

• Terms predictors and risk factors are used interchangeability in epidemiological studies. In the current study, predictor is used preferentially (revised manuscript; page 8, lines 169-184).

7. It is unclear how the 2 primary outcomes: prevalence and risk factors were used in the regression analyses. Were they merged or separated as dependent variables?

• The primary outcomes, prevalence and risk factors, were used separately as dependent variable during the analysis of regression model (revised manuscript; page 8, lines 169-177 ) .

8. In the results, ages have been stratified into 1-5 (difference of 5) and 6-12 (difference of 7). What was the criteria for this stratification?

• The stratified range covers a broad spectrum of paediatric patients and will lead to a better understanding of the most affected group(s) within the cohort (revised manuscript; page 6, lines 103-107).

9. On Table 5, what was the rationale for having those groups as the reference groups for the regression analyses? The same applies to Table 6.

• The rationale is to compare the effect of predictors/risks on affected and reference groups (revised manuscript; Tables 5 and 6). 

10. On Table 6, on the variable: insecticide use under multivariate analysis. the OR=0.95, the 95% CI=0.95-0.95 and the p-value for that variable=0.001. Mathematically, this is not possible. There is a problem with this analysis outcome.

• Additional analysis concluded that the use of insecticides did not exert a statistically significant effect and, therefore, is no longer included in multivariate analysis. Revised multivariate analysis now uses the Poisson model of regression (revised manuscript; page 8, line 169-171). 

11. Same Table 6: multivariate analysis...the p-value for age =0225.

• Data in Table 6 was updated based upon the revised use of Poisson regression for multivariate analysis (revised manuscript; page 14) .

12. I still did not get clearly what the difference was at the analytical stage between univariate and multivariate regression analysis. Was anything adjusted for and why?

• To do the multivariate analysis we should be adjusted for all list of variables in older to explain for the associated/risk factors in different variables-risks and not single variable (revised manuscript; page 8 Lines 169-178) .

Reviewer #2: The study aimed to determine the status of malaria and dengue fever, and the associated risk factors in children living in Mwanza, Tanzania. The prevalence of malaria in 436 children included in the final analysis was 15.6%, 8.5%, and 12.1% determined by MRDT, blood smear examination, and PCR, respectively. The prevalence of dengue fever determined by the NS1-RDT was 7.8%.

The study's strength was that molecular diagnosis using Polymerase Chain Reaction was incorporated into the study with a higher sensitivity. However, there are some issues with the manuscript that need to be addressed, which I have detailed here.

Introduction

Line 55: Give a reference.

• A reference is now provided (revised manuscript; line 53, Reference 1). 

Line 79: The sentence is not clear.

• The sentence was clarified (revised manuscript; page 3, line 75-77). 

Line 86: Reference

A reference was added (revised manuscript; page 4, lines 81-84, Reference 22). 

Line 90: You mentioned studies in Tanzania; there should be more than one reference.

• References 5 and 23 were added (revised manuscript, line 87). 

Line 94: Replace with associated factors.

• “Factors associated with” was substituted (revised manuscript, line 91). 

Methods

Line 120: selecting inpatients could lead to selection bias. These are severely ill patients and might have been diagnosed.

• The inpatients selected in tertiary hospitals followed the same inclusion criteria and were included to compare outcomes (revised manuscript; page 5, lines 112-113). 

Line 177: Prevalence OR

• Prevalence OR was updated based upon the common measures used in data analysis (revised manuscript; page 8, lines 169-178).

Line 180: Replace with associated factors.

• Terminology was replaced with “the associated factors in” (revised manuscript; page 8, line 177).

Results

Line 186: Replace with Males accounted for 56.2% and the mean age .....

• The percentages of males and females were included in the text (revised manuscript; page 10, lines 194 and 195). 

Line 189: Replace positive and negative with yes and no.

• Positive and negative were replaced with yes or no (revised manuscript; Tables 1-6).

Line 197-198: Be consistent with percentages. Please use %.

• The percentages of children who were vomiting, had diarrhea or body rash were 34%, 28% and 6%, respectively (revised manuscript; page 11, lines 206-208).

Line 204: change all the categories to yes and no.

• All categories were changed to Yes or No (revised manuscript; Tables 1-6).

Line 214: Prevalence of malaria=15.6%

Prevalence of dengue= 7.8

Coinfection = 14.7%?

The denominator should be 436 i.e. 15/436 =3.4% because you said of the 436 i

---

## [Editor Report · Decision Letter 1]

15 Aug 2024

Malaria and dengue fever in febrile children entering healthcare facilities in Mwanza, Tanzania

PONE-D-24-14812R1

Dear Dr. Kayange,

We’re pleased to inform you that your manuscript has been judged scientifically suitable for publication and will be formally accepted for publication once it meets all outstanding technical requirements.

Kind regards,

Benedikt Ley, PhD

Academic Editor

PLOS ONE
---

## [Editor Report · Acceptance letter]

21 Aug 2024

PONE-D-24-14812R1 

PLOS ONE

Dear Dr. Kayange, 

I'm pleased to inform you that your manuscript has been deemed suitable for publication in PLOS ONE. Congratulations! Your manuscript is now being handed over to our production team.

Kind regards, 

on behalf of

Dr Benedikt Ley 

Academic Editor

PLOS ONE